# Modeling Hsp70/Hsp40 interaction by multi-scale molecular simulations and coevolutionary sequence analysis

**Duccio Malinverni[1], Alfredo Jost Lopez[2], Paolo De Los Rios[1,3], Gerhard Hummer[2,4], Alessandro Barducci[5,6]***

[1]Laboratoire de Biophysique Statistique, Faculté de Sciences de Base, École Polytechnique Fédérale de Lausanne, Lausanne, Switzerland; [2]Max Planck Institute of Biophysics, Frankfurt am Main, Germany; [3]Institute of Bioengineering, School of Life Sciences, École Polytechnique Fédérale de Lausanne, Lausanne, Switzerland; [4]Institut für Biophysik, Johann Wolfgang Goethe Universität Frankfurt, Frankfurt am Main, Germany; [5]Inserm, U1054, Montpellier, France; [6]Université de Montpellier, CNRS, UMR 5048, Centre de Biochimie Structurale, Montpellier, France

**Abstract** The interaction between the Heat Shock Proteins 70 and 40 is at the core of the ATPase regulation of the chaperone machinery that maintains protein homeostasis. However, the structural details of the interaction remain elusive and contrasting models have been proposed for the transient Hsp70/Hsp40 complexes. Here we combine molecular simulations based on both coarse-grained and atomistic models with coevolutionary sequence analysis to shed light on this problem by focusing on the bacterial DnaK/DnaJ system. The integration of these complementary approaches resulted in a novel structural model that rationalizes previous experimental observations. We identify an evolutionarily conserved interaction surface formed by helix II of the DnaJ J-domain and a structurally contiguous region of DnaK, involving lobe IIA of the nucleotide binding domain, the inter-domain linker, and the $\beta$-basket of the substrate binding domain.

*For correspondence: alessandro. barducci@cbs.cnrs.fr

**Competing interests:** The authors declare that no competing interests exist.

## Introduction

The 70 kDa and 40 kDa Heat Shock Proteins (Hsp70/Hsp40) form the core of a chaperone machinery that plays essential roles in proteostasis and proteolytic pathways (*Daugaard et al., 2007*; *Hartl et al., 2011*; *Mayer, 2013*). Hsp70 chaperones, and their cochaperone partners Hsp40, are highly conserved ubiquitous proteins, present in multiple paralogs in virtually all known organisms (*Daugaard et al., 2007*; *Kampinga and Craig, 2010*). The chaperoning role of this machinery is based on the ability of Hsp70s to bind client proteins in non-native states, thereby preventing and reverting aggregation, unfolding misfolded proteins, assisting protein degradation and translocation (*De Los Rios et al., 2006*; *Proctor and Lorimer, 2011*; *Rampelt et al., 2012*; *Sharma et al., 2010*).

Members of the Hsp70 family are composed of two domains, connected by a flexible linker: the N-terminal nucleotide binding domain (NBD) binds and hydrolyzes ATP, whereas the C-terminal substrate binding domain (SBD) interacts with client proteins (*Zuiderweg et al., 2013*). The nature of the bound nucleotide induces dramatically different conformations of Hsp70: in the ADP-bound state, the two domains are mostly detached and behave almost independently (*Bertelsen et al., 2009*), whereas in the ATP-bound state, the SBD splits into two sub-domains that dock onto the NBD (*Kityk et al., 2012*; *Qi et al., 2013*). Therefore, nucleotide hydrolysis and exchange induce large-scale conformational dynamics that regulate the chaperone interaction with client proteins (*Mayer, 2013*).

Hsp40s are also called J-Domain Proteins, as they are invariantly characterized by the presence of a ~70 residue signature domain (J-domain), within a variable multi-domain architecture. This J-domain is composed of four helices (*Figure 1A*). The two central helices II and III form an antiparallel bundle, connected by a flexible loop with a highly conserved distinctive histidine-proline-aspartate (HPD) motif (*Pellecchia et al., 1996*). Several studies have indicated the essential role of the J-domain and of the HPD motif in Hsp40/Hsp70 interactions (*Greene et al., 1998*; *Mayer et al., 1999*; *Suh et al., 1998*; *Tsai and Douglas, 1996*). While the structural diversity of Hsp40s mirrors the functional versatility of this complex machinery, the common conserved J-domain is strictly necessary for enhancing ATP hydrolysis by Hsp70 (*Kampinga and Craig, 2010*). Modulation of Hsp70 ATPase activity through formation of transient Hsp70/Hsp40/client complexes regulates the chaperone affinity for client proteins (*De Los Rios and Barducci, 2014*; *Kellner et al., 2014*) and is hence essential for all its multiple cellular functions. Continuous switching between multiple conformations means that the chaperone-cochaperone interaction is intrinsically highly dynamic. Understanding the complex interplay between Hsp70 and Hsp40 at the mechanistic level is therefore a crucial task to gain a deeper functional insight into the chaperone machinery (*Mapa et al., 2010*).

Extensive experimental evidence of Hsp70/40 interactions has been accumulated over the last two decades (*Alderson et al., 2016*). Mutagenesis, surface plasmon resonance and NMR experiments have identified multiple putative interacting regions of the J-domain and Hsp70, mostly focusing on the *E. coli* DnaK/DnaJ system (*Ahmad et al., 2011*; *Genevaux et al., 2002*; *Greene et al., 1998*; *Suh et al., 1998*, *1999*). In spite of this considerable effort, the dynamic and transient nature of the Hsp70/Hsp40 complex has posed severe challenges to its structural characterization and no consensus view has yet been reached on this.

To date, the only available high-resolution structure has been obtained by means of X-ray crystallography of the NBD of bovine Hsc70 (Hsp70) covalently linked to the J-domain of bovine auxilin (Hsp40) (*Jiang et al., 2007*). However, this structure cannot be easily reconciled with NMR and mutagenesis data collected on DnaK/DnaJ, thus suggesting either major differences in the binding modes of bacterial and eukaryotic Hps70/40s or the trapping of a sparsely populated state that is influenced by non-native contacts (*Sousa et al., 2012*; *Zuiderweg and Ahmad, 2012*). More recently, solution PRE-NMR experiments identified an alternative highly dynamic interface between ADP-bound DnaK and DnaJ (*Ahmad et al., 2011*).

Here we relied on both multi-scale molecular modeling and statistical analysis of protein sequences to shed light on the Hsp70/Hsp40 interactions. By combining these complementary techniques, we propose a structural model of the binding of bacterial DnaK/DnaJ, which is in good agreement with available experimental data and greatly extends our understanding of this elusive yet fundamental process.

## Results

### Coarse-grained simulations identify binding regions and suggest structural models for the DnaK/DnaJ complex

We characterized DnaK/DnaJ interactions by means of Monte Carlo simulations based on a coarse-grained (CG) potential energy validated against structural and thermodynamic properties of protein complexes with low binding affinity (*Kim and Hummer, 2008*; *Kim et al., 2008*; *Różycki et al., 2011*). Binding partners are modeled as rigid bodies, using one interaction site per amino-acid (residue) located at the $C_\alpha$ position of the experimental structure. Intermolecular energy functions are based on statistical contact potentials and long-range Debye-Hückel electrostatic interactions (*Kim and Hummer, 2008*). A replica exchange Monte Carlo simulation protocol is adopted to exhaustively sample all relevant bound conformations. We took advantage of this approach and of the availability of high-resolution structures of the individual binding partners to investigate complexes formed by the J-domain of *E. coli* DnaJ (JD) with the DnaK NBD, both in its ADP- and ATP-bound conformations (NBD(ADP), NBD(ATP)). Moreover, we extended this analysis to full-length ATP-bound DnaK (FL(ATP)) in order to unveil a possible role of the SBD in the binding process.

CG trajectories were analyzed to determine the binding affinity of the three DnaK constructs for JD and to characterize the most favorable complex conformations. Calculated binding affinities ($K_D$ = 540 μM ± 60 NBD(ADP), $K_D$ = 370 μM ± 35 NBD(ATP), $K_D$ = 23 μM ± 3 FL(ATP)) are compatible

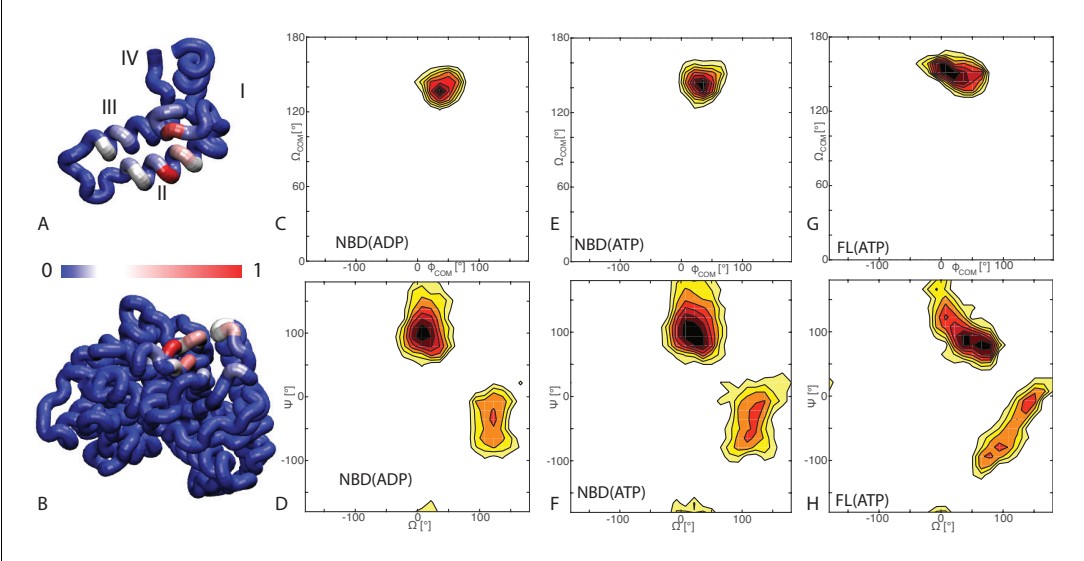

**Figure 1.** Binding modes of DnaK/DnaJ. (**A**) Probabilities of JD residues to be in contact with the DnaK (NBD(ATP) case). Helices I-IV of the JD are highlighted. See (*Figure 1—figure supplement 1*) for the NBD(ADP) and FL(ATP) cases. (**B**) Probabilities of DnaK residues to be in contact with the JD (NBD(ATP) case). Blue: low, Red: high, see scalebar. (**C,E,G**) Orientational free energy as a function of the spherical polar angles ($\Phi_{com}, \Omega_{com}$) of the JD center of mass for (**A**) NBD(ADP), (**C**) NBD(ATP), (**E**) FL(ATP). The origin and the reference axes are defined by NBD center of mass and inertia axes, respectively. (**D,F,H**) The free energy surface as a function of Euler angles ($\Omega, \Psi$) defining the relative orientation of J-domain w.r.t. the NBD for (**B**) NBD (ADP), (**D**) NBD(ATP), (**F**) FL(ATP). See Materials and methods and *Figure 1—figure supplement 4* for details on the angular definitions. Iso-lines are drawn at 1 $k_B T$ free-energy intervals. See (*Figure 1—figure supplement 2*) for the third Euler angle.

The following figure supplements are available for figure 1:

**Figure supplement 1.** Contact frequencies predicted by coarse-grained simulations.

**Figure supplement 2.** Free energy surface of bound CG conformations, using the third Euler angle.

**Figure supplement 3.** Ensemble of bound conformations predicted by coarse-grained simulations.

**Figure supplement 4.** Inertia axes of the NBD and the JD and Euler angle definition.

with previous experimental determinations (*Ahmad et al., 2011*; *Greene et al., 1998*; *Wittung-Stafshede et al., 2003*), and their significant dependence on the presence of the SBD and linker suggests a stabilizing role of this region in the DnaK/JD complex. The analysis of the conformational ensembles corresponding to bound complexes revealed several distinctive features of the DnaK/JD binding process, along with a certain degree of conformational heterogeneity (*Figure 1—figure supplement 3*). The free energy surfaces as a function of NBD-centered spherical coordinates (*Figure 1C,E,G*) clearly indicate that a specific binding site predominates in all the simulated DnaK constructs, irrespective of the bound nucleotide and of the presence of the SBD and linker.

To better characterize this favored binding interface, we calculated the probability of each DnaK residue to be in direct contact with JD and mapped it onto the NBD structure (*Figure 1B*). These results suggest that the formation of DnaK/DnaJ complexes mostly involves a DnaK region located on lobe IIA of the NBD, in a negatively charged narrow groove formed by a beta-sheet and a short loop (*Figures 1B* and *2A,B*). The complementary interface on JD suggests that its interaction with DnaK is mostly mediated by the positively charged helix II and few residues on helix I (*Figure 1A*).

We further analyzed the conformational ensembles obtained by CG simulations to identify the most relevant orientations of the JD in the bound complexes. The free energy surfaces as a function of Euler angles measuring the relative orientation of the binding partners reveal two major conformational sub-ensembles in all the simulated systems (*Figure 1D,F,H*, *Figure 1—figure supplement 2*). Cluster analysis of the CG trajectories suggests that these distinct intermolecular arrangements

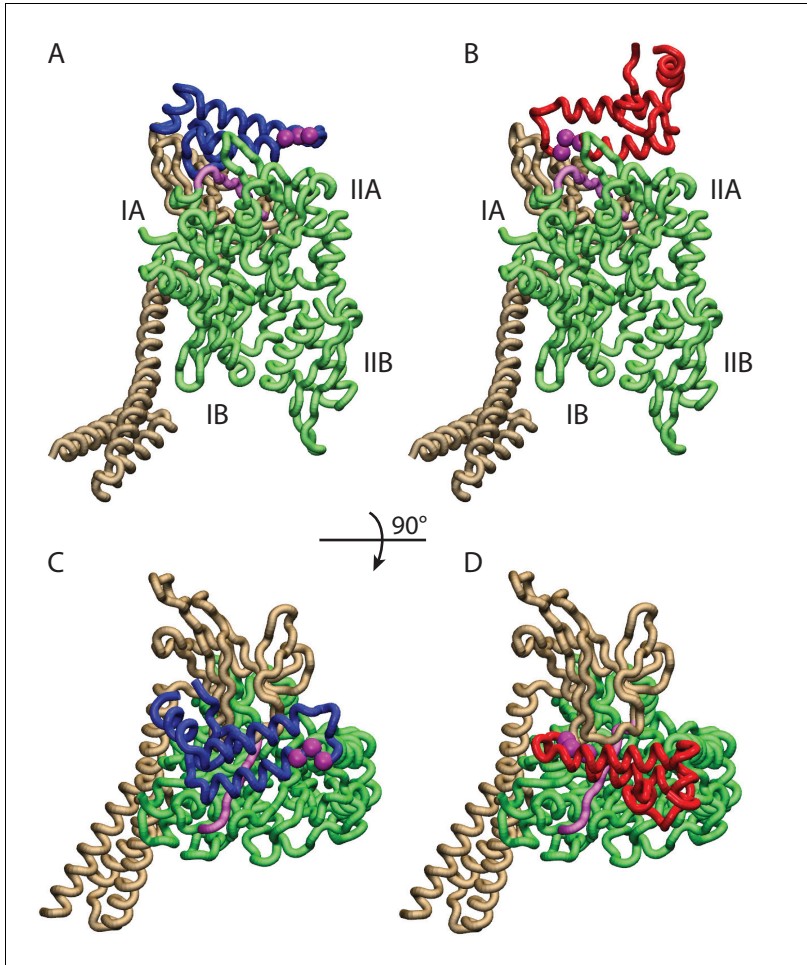

**Figure 2.** Conformations of the DnaK:JD complex. Most representative HPD-OUT/IN conformations for the FL (ATP):JD system (HPD-OUT: (A) and (C), HPD-IN: (B) and (D)). The four lobes forming the sub-structures of the NBD are highlighted. The SBD is in brown. The docked inter-domain linker is in purple. The HPD tripeptide of the J-domain is depicted as magenta spheres. The JD is depicted in blue (HPD-OUT) or red (HPD-IN). See (*Figure 2—figure supplement 1*) for the NBD(ADP) and NBD(ATP) systems. For readability, the NBD in the rotated panels C) and D) is uniformly colored in green.

The following figure supplement is available for figure 2:

**Figure supplement 1.** HPD-IN/OUT conformations for NBD(ADP) and NBD(ATP).

---

correspond to complexes with similar binding interfaces but opposite orientations of the JD with respect to DnaK (*Figure 2*, *Figure 2—figure supplement 1*). The conserved HPD loop of the JD points outwards in one conformational sub-ensemble (HPD-OUT, *Figure 2A,C*), whereas in the other it is close to the groove on the NBD where the inter-domain linker docks (HPD-IN, *Figure 2B,D*). These two arrangements were observed in all the simulated systems with very limited perturbations because of the presence of SBD, and together they account for more than 91% of the populations in the bound ensembles. In all the simulated systems, the population of the HPD-OUT conformation is higher than that of HPD-IN. However, the observed free energy differences between the two binding modes (~1.5 kcal/mol) are comparable with the expected uncertainty of the CG model (*Kim and Hummer, 2008*).

## Coevolutionary analysis predicts conserved DnaK-DnaJ contacts

Statistical analysis of the covariation in multiple sequence alignments (MSAs) represents an extremely valuable approach to investigate protein structure by identifying residue-residue interactions that are evolutionarily conserved (*de Juan et al., 2013*; *Marks et al., 2012*). Particularly, direct coupling analysis (DCA) (*Morcos et al., 2011*; *Weigt et al., 2009*) of paired MSAs of interacting proteins has been successfully applied to predict interfaces of protein complexes (*Hopf et al., 2014*; *Ovchinnikov et al., 2014*). The canonical matching algorithms for generating paired MSAs are based on intergenic distances and, unfortunately, they cannot be directly applied to the Hsp70/40 interaction, because of promiscuous interactions that cannot strictly be predicted by operon structure in this family. To circumvent this difficulty, we adopted an alternative approach based on generation of an ensemble of stochastically matched MSAs. In this context, the statistical reliability of inter-residue couplings can be related to their frequency of appearance within the DCA predictions obtained from all the realizations of the MSA ensemble (see Materials and methods for details).

We took advantage of the large sizes of the Hsp70/40 families (Hsp70: 20061 sequences, Hsp40: 26254 sequences, distributed in all kingdoms, see Materials and methods) to evaluate inter-residue evolutionary couplings between the Hsp40 JD and the Hsp70 NBD. The high degree of conservation of these domains guarantees high-quality alignments and thus accurate results. This analysis identified three inter-protein residue pairs that stand out among coevolving pairs in the Hsp40 and Hsp70 families (*Figure 3A*), corresponding to N187-K23, D208-K26 and T189-R19 in *E. coli* DnaK and DnaJ. The spatial proximity of N187/D208/T189 on the DnaK NBD, and the proximity between K23/K26/R19 on the JD, suggest the presence of well-defined evolutionarily conserved binding patches across Hsp40/70 families.

Remarkably, these patches are perfectly overlapping with the binding regions predicted by CG modeling, that is, helix II in DnaJ and a sub-region of lobe IIA in DnaK NBD. Thus, DCA predictions can be used to evaluate the HPD-IN and HPD-OUT binding modes suggested by CG simulations (*Figure 3B,C*). Quantitative assessment is limited by several factors such as the difficulty of translating coevolutionary couplings into exact distance restraints, the limited resolution of the residue-based CG model and the dynamic nature of the JD/DnaK complexes. Nevertheless, while both the intermolecular conformations might be compatible with DCA predictions, the specific binding pattern predicted by coevolutionary analysis matches significantly better the JD orientation observed in HPD-IN (*Figure 3C*). Moreover in this conformation, the location of D35 in the HPD motif is compatible with a putative interaction with R167 on DnaK NBD, as previously suggested by mutagenesis experiments (*Suh et al., 1998*). The overall better agreement of the HPD-IN conformation can be further strengthened by the observation that the average $C_\alpha - C_\alpha$ distance observed in HPD-OUT for the pair T189-R19 seems too large ($>20 Å$) to justify a direct interaction, and thus a strong statistical coupling between those residues.

We repeated the DCA analysis on two subsets restricted to either bacterial or eukaryotic sequences, to further investigate the origin of the detected coevolutionary signal. The results obtained on the bacterial subset (*Figure 3—figure supplement 1A*) are in perfect agreement with the results obtained on the full dataset, whereas no strong coevolutionary couplings are detected using the eukaryotic subset (*Figure 3—figure supplement 1B*). This observation indicates that the coevolutionary signal of the observed Hsp70/Hsp40 interface mostly originates from the bacterial sequences in the dataset.

## Atomistic MD simulations highlight a dynamical interface

We performed atomistic explicit-solvent molecular dynamics (MD) simulations to investigate the stability and the dynamics of the DnaK/DnaJ docked conformations obtained with CG modeling.

Firstly, we assessed the reliability of the DnaK/DnaJ complexes by performing 10 MD runs of 30 ns for each system (JD:NBD(ADP), JD:NBD(ATP) and JD:FL(ATP)) in both HPD-IN and HPD-OUT binding modes (see Materials and methods). These simulations showed a certain degree of conformational dynamics in all cases (*Figure 4—figure supplement 1*), and provided information about the relative stability of the various complexes. Particularly, the $C_\alpha$ distance root mean square deviation (dRMS) and the average angular deviation $\Theta$ of the JD with respect to the starting frame (*Table 1*, *Figure 4—figure supplements 2–3*) revealed that NBD(ADP):JD and NBD(ATP):JD complexes in the HPD-OUT conformations displayed high structural variability in contrast with HPD-IN

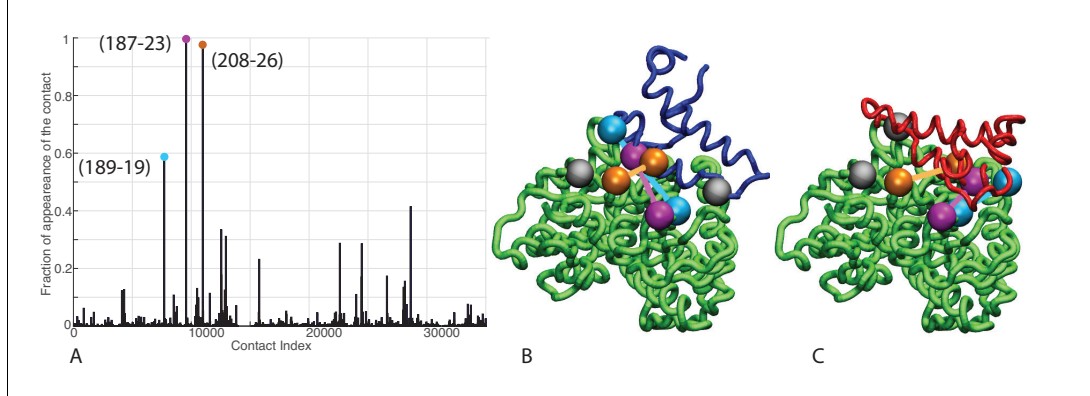

**Figure 3.** Contacts from coevolutionary analysis. (**A**) Frequency of appearance of the coevolutionary inter-protein contacts (see Materials and methods). The three most frequent contacts are highlighted (N187-K23 magenta, D208-K26 orange and T189-R19 blue. Numbering refers to the *E. coli* DnaK-DnaJ (Uniprot IDs: DnaK P0A6Y8, DnaJ P08622)). (**B–C**) The same three contacts represented on the HPD-OUT (blue, panel B) and HPD-IN (red, panel C) conformations of the NBD(ATP):JD complex. Coevolving residues are depicted by spheres, following the color scheme of panel A. Gray spheres represent D35 of the 33HPD35 motif on JD and R167 of the NBD. (See Materials and methods and *Figure 7* for an extended DCA analysis and validation.)

The following figure supplement is available for figure 3:

**Figure supplement 1.** Frequency of appearance of the coevolutionary inter-protein contacts (see Materials and methods) on separated bacterial and eukaryotic datasets.

conformations, which were significantly more stable on this time scale. This difference was significantly less pronounced in the simulations of full-length DnaK, likely because of the stabilizing effect of JD-SBD interactions.

We then focused on the FL(ATP):JD complex in the HPD-IN conformation, which stands out among the other systems as it involves full-length DnaK and is compatible with coevolutionary and mutagenesis data. Particularly, we performed three MD simulations of 1 $\mu$s to better probe its conformational dynamics. The results confirmed the stability of the HPD-IN arrangement on a more extended time scale but unveiled the presence of multiple, distinct conformational states within this overall binding mode (*Figure 4A*). While an exhaustive characterization of the conformational space exceeds the capabilities of all-atom MD, the broad structural ensembles are suggestive of a significant degree of conformational dynamics in the $\mu s$ timescale. This picture is consistent with the broad, multi-modal distributions of the atomic distances corresponding to relevant intermolecular interactions, such as the three evolutionarily conserved contacts and D35-R167 (*Figure 4B*). These interactions thus appear to be transiently populated in the context of a highly dynamical intermolecular interface.

**Table 1.** Stability analysis of atomistic MD. dRMS is the distance root mean square deviation between $C_\alpha$ atoms of the JD and DnaK. $\Theta$ is the angle formed by the principal axis of the JD with respect to the principal axis of the JD in the starting frame. Standard deviations over the 10 MD trajectories are reported in parentheses. See Materials and methods for details.

|  |  | dRMS [Å] | $\Theta$ [°] |
|---|---|---|---|
| NBD(ADP) | HPD-OUT | 9.2(2.1) | 46.9(15.8) |
|  | HPD-IN | 3.7(0.9) | 15.6(6.1) |
| NBD(ATP) | HPD-OUT | 12.3(5.0) | 40.9(17.4) |
|  | HPD-IN | 6.1(3.0) | 19.9(12.9) |
| FL(ATP) | HPD-OUT | 6.2(2.7) | 18.6(8.5) |
|  | HPD-IN | 5.2(2.0) | 21.8(5.1) |

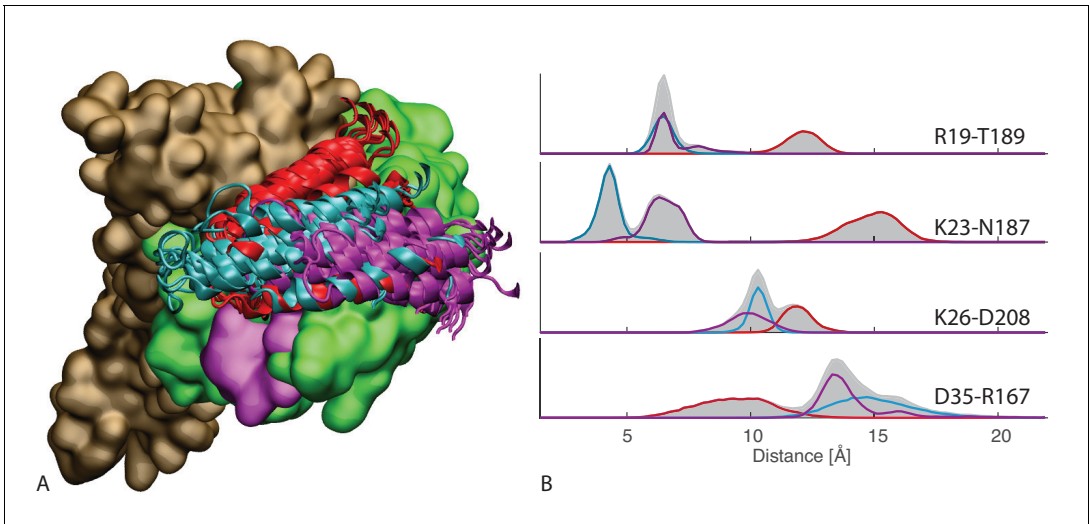

**Figure 4.** All-atom $1\mu s$ simulations of FL-ATP. (**A**) 10 snapshots of the three long atomistic MD simulations of FL-ATP DnaK bound to the DnaJ JD. Green: NBD, magenta: linker, brown SBD, red/cyan/purple: JD for the three trajectories. For ease of visualization, only helices II and III of the JD are depicted. (**B**) Distributions of the distances of the three coevolving contacts and the D35-R167 contact. The distance distributions for the three cases follow the color scheme of panel A (red/cyan/purple). Shaded areas are the sum of the distributions for each case. See (*Figure 4—figure supplement 4*) for traces of the trajectories.

The following figure supplements are available for figure 4:

**Figure supplement 1.** Atomistic stability analysis.

**Figure supplement 2.** dRMS of atomistic MD trajectories.

**Figure supplement 3.** Angular deviation of atomistic MD trajectories w.r.t. the central HPD-IN/OUT CG conformations.

**Figure supplement 4.** dRMS and angular deviations of the $\mu s$ atomistic simulations.

To shed further light on the molecular determinants of the DnaK/JD interaction, we then evaluated the energetic contributions of individual residues to the protein-protein binding energy in the $1\mu s$ trajectories, using a generalized born surface area (GBSA) approximation (see Material and Materials and methods). The per-residue decomposition of the binding energy highlighted four fragments of DnaK that contribute most strongly to the stabilization of the DnaK/JD complexes (*Figure 5A*). The residues corresponding to three of these spots form an almost continuous patch covering the upper-cleft between lobes II and III of the NBD (residues 206–219, 329–335) and a segment of the inter-domain linker (residues 391–393) *Figure 5A–B*). Reciprocally, the energetic analysis of the JD residues predicted helix II and the HPD loop as being the principal region involved in energetic stabilization of the complex (*Figure 5—figure supplement 1*). These findings confirm that the JD strongly interacts with the docked linker and its neighboring residues, suggesting the role of JD at stabilizing this linker arrangement. Remarkably, the energetic analysis also unveiled that a stretch of the SBD beta-basket plays an important role in securing the DnaK/JD interface (residues 414–423).

To investigate the functional relevance of the interaction between the JD and the SBD, we then extended the DCA analysis to full-length Hsp70 sequences. This analysis confirmed the significance of the previously observed NBD/JD contacts and predicted two inter-protein contacts involving the SBD (H422-E75 and Q424-K51 in *E. coli* DnaK/DnaJ numbering, *Figure 5C* and *Figure 5—figure supplement 2*). Interestingly, the two residues on the SBD correspond to the SBD region energetically involved in the binding of the JD, and thus show that their interaction has been conserved through evolution. Of the two corresponding residues on the JD, one is located on helix III (K51), in excellent agreement with the HPD-IN binding mode, while the second (E74) lies in the unstructured

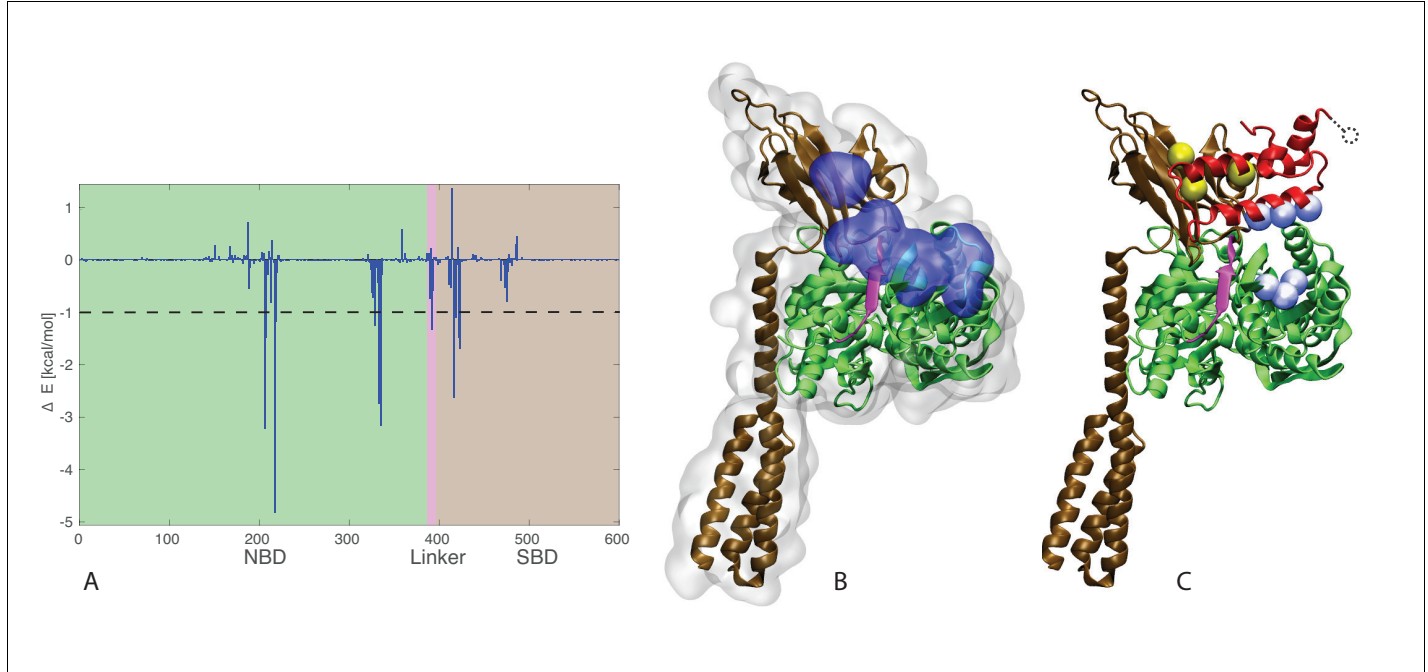

**Figure 5.** J-domain - SBD interactions. (A) DnaK per-residue contribution to the binding energy with the J-domain. The NBD, linker, and SBD regions are highlighted in green, pink, and ochre, respectively. The dashed line denotes the threshold for which residues are depicted in panel B (−1kcal/mol). (B) Structural view of the residues most contributing to the binding energy with the J-domain. The subdomains of DnaK (NBD, linker, and SBD) follow the same color scheme as in panel A. Residues significantly contributing to the binding energy ($\Delta E < -1$ kcal/mol) are depicted in blue surface representation. (C) Coevolutionary contacts predicted on the full-length Hsp70 sequences. The three blue contacts are the same as those reported in *Figure 3*. The two contacts involving the SBD of Hsp70 are shown in yellow. The dotted circle represents residue E75 of the J-domain, absent from the structures used in the simulations (see Materials and methods). The depicted conformation is the final frame of one of the three $1 - \mu s$ HPD-IN FL-ATP simulations.

The following figure supplements are available for figure 5:

**Figure supplement 1.** J-domain per-residue contribution to the binding energy with DnaK.

**Figure supplement 2.** Frequency of appearance of the coevolutionary inter-protein contacts (see Materials and methods), for the dataset containing full-length Hsp70s.

C-terminal region of the JD, which was not included in the structural model (see Materials and methods). Taken together, these energetic and evolutionary analyses strongly indicate that the J-domain directly interacts with the inter-domain linker in its docked conformation, as well as with the SBD.

## Discussion

The integration of complementary approaches such as coevolutionary sequence analysis and molecular modeling at the coarse-grained and atomistic scale allowed us to shed light on the structural details of the crucial interaction of DnaJ with DnaK.

Indeed, molecular simulations based on a CG model specifically suited to study low-affinity protein binding identified the positively charged helix II of the JD and a region close to lobe IIA of the DnaK NBD as the most relevant interaction sites in the formation of DnaK/JD complexes. This prediction was corroborated by statistical sequence analysis showing that several inter-protein contacts across this interface strongly coevolve in the Hsp40/70 family. These findings are in good agreement with much experimental evidence collected in the last 20 years. Indeed, a major role for helix II of JD in the DnaK/DnaJ interaction has been suggested both by NMR and mutagenesis experiments (*Greene et al., 1998*; *Suh et al., 1998*). Furthermore, our prediction is in excellent agreement with recent PRE-NMR investigation of the interaction of JD with ADP-bound DnaK that identified the

sequence [206]EIDEVDGEKTFEVLAT[221] as the main binding region on DnaK (*Ahmad et al., 2011*). The observation that the same interaction site was present in all the simulated systems (ATP- and ADP-NBD and ATP-bound full-length DnaK) strongly suggests that this region is likely to play a primary role throughout the chaperone functional cycle, thus greatly extending its physiological relevance. Interestingly, the predicted bound conformations located the J-domain in near proximity to the docked inter-domain linker in FL(ATP) (*Figure 2*), which has been shown to play a central role in the allosteric coupling of the two domains in the Hsp70 cycle (*Alderson et al., 2014*; *Vogel et al., 2006*; *Zhuravleva et al., 2012*).

Beyond a detailed characterization of the binding regions on DnaK and the JD, our integrated approach provided precious information about the inter-protein arrangement in the transient DnaK/JD complexes. Effectively, CG modeling suggested two possible binding modes characterized by opposite orientations of the JD (*Figure 2*). Both these putative conformations were only minimally affected by structural differences in the NBD upon ATP/ADP binding or by inter-domain docking in full-length ATP-bound DnaK. Direct comparison of these results with the interaction pattern inferred from coevolutionary analysis reveals an excellent agreement for one of the conformations (HPD-IN). Further elements supporting the relevance of this structure can be found by taking into account the role of the highly conserved HPD loop of the JD. Indeed, several mutagenesis studies have shown that the HPD loop is fundamental for functional chaperone/cochaperone interactions (*Landry, 2003*; *Suh et al., 1998*). NMR investigations have reported conflicting evidence about the actual involvement of the HPD region in the Hsp70/Hsp40 interface (*Ahmad et al., 2011*; *Greene et al., 1998*; *Kim et al., 2014*). However, the observation that the DnaK R167H mutation could suppress the deleterious effect of the DnaJ D35N mutation strongly pointed toward a direct, yet possibly transient, interaction of these residues during the chaperone functional cycle (*Suh et al., 1998*). Strikingly, this experimental evidence is perfectly compatible with the spatial proximity of DnaJ D35 and DnaK R167 in the HPD-IN conformation (*Figure 3C*), whereas it cannot be easily reconciled with the orientation of the HPD in the current structural model of the DnaK:JD complex based on PRE-NMR experiments (*Ahmad et al., 2011*). The HPD-IN conformation hence provides a novel, suggestive model for the elusive DnaK/DnaJ complex that best recapitulates the most relevant experimental evidence on prokaryotic Hsp70/Hsp40 systems.

The insights obtained combining CG modeling and coevolutionary sequence analysis were further confirmed and enriched by explicit solvent, atomistic simulations. Indeed, MD trajectories confirmed the overall stability of the HPD-IN complex on the $\mu s$ time scale and showed the transient interaction of D35-R167 and of the coevolving pairs (*Figure 4*). The transient nature of these contacts is perfectly compatible with the dynamical interface suggested by NMR experiments (*Ahmad et al., 2011*). Furthermore, energetic analysis of the atomistic simulations unveiled the residues that contribute most to the formation of this dynamical complex. Particularly, we notice that the interaction of JD helix II with the DnaK surface composed of the docked intermolecular linker and adjacent $\beta$-strands has a key role in stabilization of the binding interface. Remarkably, the MD analysis highlighted that a few residues of the SBD $\beta$-basket contribute significantly to JD binding. DCA analysis performed with full-length Hsp70 sequences further strengthened this observation by showing that the SBD/JD interface observed in the simulations actually contains pairs of coevolving residues in the Hsp70/Hsp40 family. Altogether, our results suggest that although the overall DnaK/JD arrangement is determined by interactions with the NBD, specific contacts with both the SBD and the inter-domain linker may significantly increase the complex stability, thus rationalizing the decreased affinity observed for isolated NBD (*Kim et al., 2014*).

To put our findings into context, we have to take into account the current understanding of allosteric signal transmission in DnaK. Much experimental evidence has indicated that ATP-bound DnaK undergoes large-scale structural fluctuations with significant inter-domain rearrangements (*Mayer, 2010*; *Mapa et al., 2010*). Within this conformational ensemble, NMR and mutagenesis studies suggested that allosterically active conformers with high ATPase activity are characterized by a docked inter-domain linker but very limited SBD-NBD contacts (*Zhuravleva et al., 2012*; *Kityk et al., 2015*; *Jiang et al., 2007*). Remarkably, we find that the docked inter-domain linker and neighboring residues correspond to a hotspot for Hsp70/Hsp40 interaction, suggesting a stabilization of this linker conformation in the transient DnaK/JD complex. Furthermore, the energetically favorable and evolutionarily conserved interactions between DnaK SBD and JD suggest an additional mechanism for altering the conformational ensemble of ATP-bound DnaK upon DnaJ binding. Our

structural model is thus compatible with the intriguing hypothesis that the docking of the JD affects the SBD dynamics through direct interactions, shifting DnaK towards an allosterically active conformation (*Zhuravleva et al., 2012*). Therefore, while the dynamical interplay among NBD, interdomain linker, SBD, and JD remains to be fully elucidated, our analysis provides insights about the regulatory role of J-domain proteins in the Hsp70 cycle, a topic of great interest for understanding the role of the Hsp70 machinery in the global chaperone network (*Kravats et al., 2017*) as well as for designing allosteric inhibitors (*Li et al., 2016*).

The alternative arrangement observed in the bovine auxilin:Hsc70 complex (*Jiang et al., 2007*), and its poor agreement with NMR/mutagenesis data on DnaK/DnaJ (*Ahmad et al., 2011*; *Greene et al., 1998*) raise the question of the uniqueness of the Hsp70/Hsp40 binding mode (*Garimella et al., 2006*). Whether these major structural differences are caused by artifacts introduced by the artificial cross-linking, or point to the existence of multiple dynamic interaction interfaces, or to phylogenetic differentiation of Hsp70/40s, remains an essential yet unsolved question. In this respect, the successful combination of coevolutionary and molecular modeling analysis proposed here paves the way for further analysis to tackle these challenges.

## Materials and methods

### Coarse-grained simulations

#### Simulation protocol

We used the coarse-grained model introduced in *Kim and Hummer (2008)* to simulate the binding of DnaJ to DnaK constructs. Both proteins were treated as rigid bodies, at a resolution of one bead per residue centered on the $C_\alpha$ atoms. We modeled NBD(ADP) using the structured region (residues 4–380) of ADP-bound *E. coli* DnaK (pdb: 2kho [*Bertelsen et al., 2009*]), whereas we relied on the X-ray structure of ATP-bound *E. coli* DnaK (pdb: 4jne [*Qi et al., 2013*]) for both NBD(ATP) (residues 4–380) and FL(ATP) (residues 1–600). The J-domain was modeled based on the structured region of the *E. coli* DnaJ (pdb: 1xbl [*Pellecchia et al., 1996*], residues 2–70). We defined the structured part of the J-domain, by aligning multiple J-domains (pdb:1xbl, 4j7z, 2m6y, 2n04, 2qsa, 2lgw, 2och, 2dn9, 2dmx, 1hdj, 1faf, 2ctw) and considering the common structured part. We therefore removed the last six C-terminal residues from the 1xbl structure to define the maximal common structured region of J-domain. Our definition of the J-domain corresponds to the one used in *Ahmad et al. (2011)*.

Conformations were sampled from the equilibrium distribution using a replica-exchange Monte-Carlo (REMC) algorithm in a prediodic box, with 20 replicas distributed in the temperature range 200–395K. A total of $2 \cdot 10^6$ MC-steps were performed for each replica and samples were recorded every 100 steps. Dissociation constants were calculated by measuring the fraction of bound conformations, and simulations were repeated with five increasing box sizes (240–360 Å for NBD, 300–420 Å for FL(ATP)). Bound conformations were extracted by selecting all complexes in which the two proteins had at least one pair of beads within 8 Å distance and total interaction energy equal or below $-2k_BT$. All subsequent analyses on the CG complexes have been performed on the ensemble of bound complexes. The algorithm introduced in (*Daura et al., 1999*) with a cutoff radius of 5 Å was used to perform cluster analysis of the CG trajectories.

#### Angular analysis

To characterize the angular orientation of the binary complexes, we used two sets of angular coordinates. The binding site of the JD on the DnaK NBD was first characterized by the spherical coordinates of its center of mass. Let $\vec{J}_{CM}$ and $\vec{K}_{CM}$ denote the center of mass of the JD and NBD, respectively (computed over all $C_\alpha$ atoms). Furthermore, let $\vec{I}_i^J$ and $\vec{I}_i^K$ ($i = 1, 2, 3$, $i = 1$ corresponds to the largest moment of inertia) denote the three (normalized) axes of inertia of the JD and NBD, respectively. The spherical coordinates $\Phi_{COM}, \Omega_{COM}$ describing the binding site of the JD on the NBD are then defined by the usual pair of spherical angles, in the reference coordinate system defined by the inertia axis of the NBD.

The relative orientation of the JD with respect to the NBD is characterized by three Euler angles $\Theta, \Omega, \Psi$, computed with respect to the reference frame of the NBD, as follows (see *Figure 1—figure supplement 4*):

$$\Theta = \frac{180°}{\pi} \operatorname{asin}\left(\vec{I}_1^J \cdot \vec{I}_2^K\right)$$

$$\Omega = \frac{180°}{\pi} \operatorname{atan2}\left(\frac{\vec{I}_2^J \cdot \vec{I}_2^K}{\cos(\Theta)}, \frac{-\vec{I}_3^J \cdot \vec{I}_2^K}{\cos(\Theta)}\right)$$

$$\Psi = \frac{180°}{\pi} \operatorname{atan2}\left(\frac{\vec{I}_1^J \cdot \vec{I}_3^K}{\cos(\Theta)}, \frac{\vec{I}_1^J \cdot \vec{I}_1^K}{\cos(\Theta)}\right)$$

where $\operatorname{atan2}$ denotes the quadrant-checking arctangent function.

## Coevolutionary analysis

### Sequence extraction and preprocessing

To perform direct-coupling analysis we used the same sequence extraction protocol as in (*Malinverni et al., 2015*), reported hereafter:

- • Initial seeds were built for both protein families, containing sequences from all kingdoms: Bacteria, Eukaryotes and Archaea.
- • Hidden Markov models of the alignments were built, using the hmmbuild utility of the HMMER (version 3.1b2) (*Mistry et al., 2013*) suite, with default parameters.
- • The union of the Swissprot and the Trembl databases (release 2015_08) was scanned against these two profiles , using the hmmsearch (*Mistry et al., 2013*) utility, with default parameters.
- • For both retrieved MSAs, all sequences having more than 10% gapped positions were removed from the datasets.
- • The Hsp70 sequences were restricted to the NBD and linker region, by trimming the C-terminal part of the alignment.
- • Taxonomic identifiers for all sequences were retrieved from the NCBI taxonomy database.

This resulted in multiple-sequence-alignments for the Hsp70 and Hsp40 families containing, respectively, 20061 and 26254 homologues, with the following taxonomic distribution:

The number of paralogs per organism varies widely (*Table 2*, *Figure 6*), with bacteria having fewer paralogs for both Hsp40 and Hsp70. All organisms were included in the full dataset, both those having multiple paralogs and organisms possessing a single copy of Hsp40-Hsp70 pairs (3701 organisms).

The resulting MSAs covered the following ranges of the *E. coli* DnaK/DnaJ proteins: DnaK (Uniprot ID: P0A6Y8) I4-T395, DnaJ (Uniprot ID: P08622) K3-G78. Both sequence alignments and taxonomic identifiers for the Hsp40 and Hsp70 families are available as supplementary material (*Supplementary file 1−5*).

### Direct-coupling analysis

Direct-coupling analysis (DCA) was performed on each of the 1000 stochastically concatenated MSAs using the asymmetric version of the pseudo-likelihood method (*Ekeberg et al., 2014*), with standard parameters (maximum 90% sequence identity, regularization parameters $\lambda_H = \lambda_J = 0.01$). In practice, the parameters $J_{ij}(A,B)$ and $h_i(A)$ of the generalized Potts model (*Equation 1*) are

**Table 2.** Summary of the taxonomic composition of the Hsp40 and Hsp70 alignments. Entries of the table represent the number of sequences found in each taxonomic group. The number of organisms in each taxonomic group is indicated in parenthesis.

| | Eukaryotes | Bacteria | Archaea | Viruses | Other | Total |
|---|---|---|---|---|---|---|
| Hsp40 | 14369 (1093) | 11379 (7837) | 311 (273) | 36 (22) | 159 (13) | 26254 (9238) |
| Hsp70 | 7881 (1933) | 11819 (8272) | 273 (258) | 25 (17) | 63 (13) | 20061 (10493) |

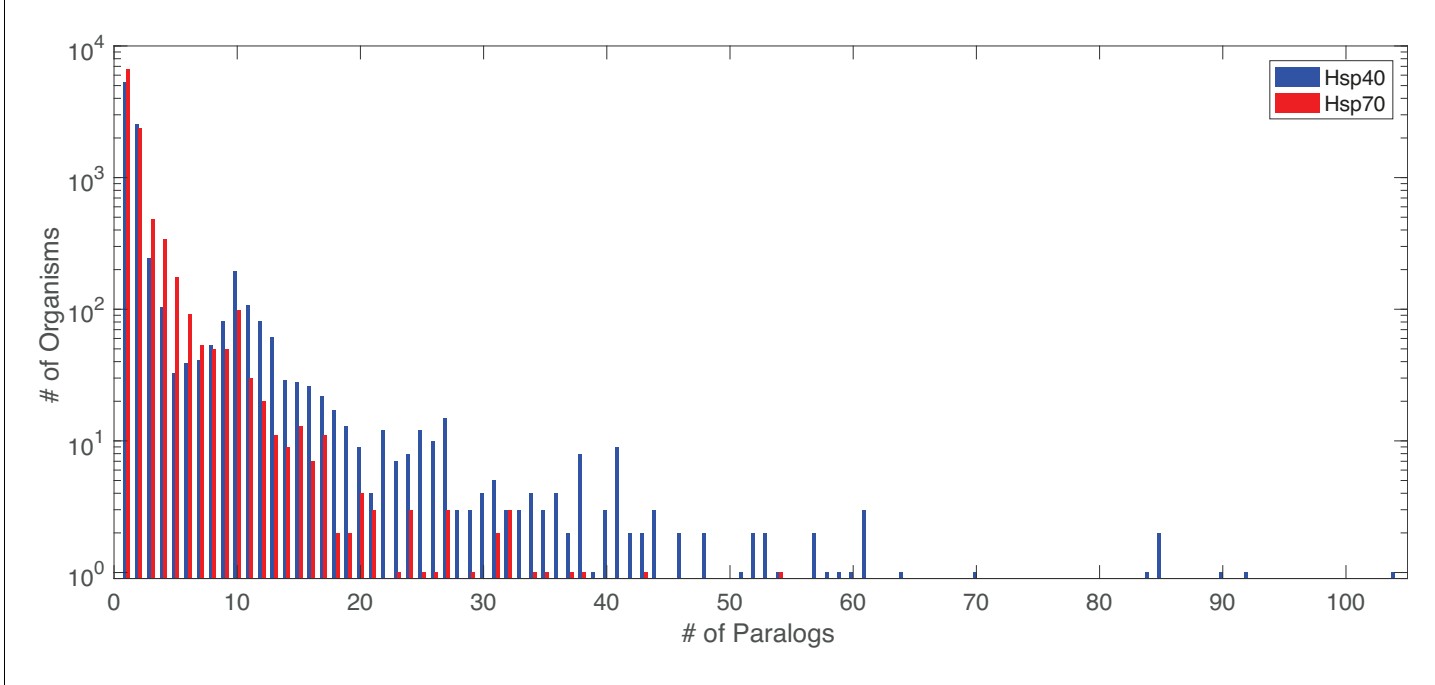

**Figure 6.** Distribution of number of paralogs per organism, for Hsp40 and Hsp70.

numerically fitted to the data in the MSAs in the Pseudo-Likelihood approximation (*Ekeberg et al., 2014*)

$$P(\mathbf{X}) = \frac{1}{Z}\exp\left(\sum_{ij} J_{ij}(X_i, X_j) + \sum_i h_i(X_i)\right) \quad (1)$$

where $\mathbf{X}$ denotes an amino-acid sequence, $Z$ the normalizing partition function. The raw DCA scores $S_{ij}^{raw}$ (*Equation 2*), quantifying the statistical coupling strength between two positions in the MSA, are defined as the Frobenius norm of the local $21 \times 21$ coupling matrices $J_{ij}(A, B)$

$$S_{ij}^{raw} = \sqrt{\sum_{A,B} J_{ij}(A, B)^2} \quad (2)$$

Finally, we apply the average product correction (APC) (*Dunn et al., 2008*) (*Equation 3*) to the raw DCA scores, to correct for a bias in position specific mutation rate. To account for variable mutation rates in two different protein families, we follow the modification to the APC introduced in (*Ovchinnikov et al., 2014*), by taking the average over the two protein segments independently

$$S_{ij} = S_{ij}^{raw} - \frac{S_{i.}^{raw} S_{.j}^{raw}}{S_{..}^{raw}} \quad (3)$$

where . denotes the average over the row/column of the matrix $S_{ij}$.

## Random paralog matching

To detect inter-protein coevolving residue pairs, concatenated MSAs of interacting protein sequence pairs must be built. Given the lack of knowledge on the interaction network of Hsp40s and Hsp70s and the lack of conservation of the number of paralogs throughout species, no trivial matching could be performed. Furthermore, the approach of matching interacting sequence pairs based on their genomic proximity (*Feinauer et al., 2016*; *Hopf et al., 2014*; *Ovchinnikov et al., 2014*) failed because of a lack of operon organization of Hsp70s and Hsp40s. We therefore employed a stochastic approach to the sequence-matching problem, which consists of the following steps:

- For each organism:

  - Randomly select a sequence of Hsp70 and randomly match it to a single Hsp40 sequence of the same organism.
  - Remove these two sequences from the pool of available sequences in the current organism.
  - Repeat this procedure until there are no more Hsp40 or Hsp70 sequences to match in the current organism.
  - Repeat the procedure for all organisms possessing at least one Hsp40 and Hsp70 sequence.

This procedure generated a stochastic realization of a matched MSA, ensuring that each sequence was present only once in the MSA. This constraint of matching each sequence only once avoided the combinatorial explosion of the size of the random MSAs and, consequently, the dilution of the coevolutionary signal because of the presence of an overwhelming majority of non-interacting protein pairs. We generated 1000 such random MSAs and performed DCA on each of them individually. For each DCA realization, we then extracted the strongest inter-protein coevolving pairs, using a criterion introduced in *Hopf et al. (2014)*, which briefly goes as follows: all inter-protein DCA scores are renormalized as

$$\tilde{S}_{ij} = \frac{S_{ij}}{|min(S_{ij}^{Inter})|\left(1 + \sqrt{\frac{N}{N_{eff}}}\right)} \tag{4}$$

where $S_{ij}$ denotes the average-product corrected DCA score, $N$ the length of the MSA and $N_{eff}$ the effective number of sequences in the MSA (taking into account the reweighting by maximum 90% identity). Note that the minimum is taken restricted to the interface scores. This renormalization partially corrects for dependencies of the inter-protein DCA scores on the alignment width ($N$) and depth $N_{eff}$, and therefore permits an easier comparison of DCA scores across different protein families (*Hopf et al., 2014*). Note that the introduction of this score does not change the relative ranking of inter-protein DCA contacts, as it is merely a convenient renormalization of the scores. For each of the 1000 realizations, we collected all the inter-protein DCA pairs which had a normalized score $\tilde{S}_{ij}$ above 0.8. We then computed the selection frequency for each contact (*Figure 3A*) and retained the residue pairs that were selected most frequently for subsequent analysis.

The rationale behind this procedure was that contacts appearing repeatedly in multiple random realizations were robust to matching noise in the MSAs and should therefore reflect a strong underlying coevolutionary signal.

As mentioned above, only a single copy of Hsp40 and Hsp70 sequences were retrieved in some organisms. The corresponding Hsp70/Hsp40 pairs were systematically matched and added to all the the randomly generated MSAs. To investigate the dependence of our results on this choice, we performed a DCA analysis on a limited MSA composed solely of single-copy organisms. We observed that in this case, only a fraction of the strongest coevolutionary signals predicted from the full-dataset are recovered.

Recently, two methods to simultaneously pair interacting paralogs and predict inter-protein coevolving contacts have been developed (*Bitbol et al., 2016*; *Gueudré et al., 2016*). Both these involved methods tackle the combined objective of deciphering the paralog interaction network as well as determining coevolving residue across protein interfaces through iterative schemes. We observed that the strongest inter-protein contacts predicted by the two methods (*Bitbol et al., 2016*; *Gueudré et al., 2016*) strongly overlap with those obtained using our random-matching strategy (*Figure 7—figure supplement 1*). To allow comparison with other inter-protein interactions studied by DCA, we report here an empirical inter-protein score introduced in (*Feinauer et al., 2016*; *Gueudré et al., 2016*), which consists of characterizing the overall strength of inter-protein coevolution by the average of the DCA scores of the four strongest inter-protein pairs (scored by the Frobenius norm and after APC correction). We obtained empirical inter-protein scores of 0.11, 0.17, and 0.24 for the random matching, PPM, and IPA, respectively. Note that in the case of the random matching, the empirical inter-protein score is averaged over the 1000 realizations of the

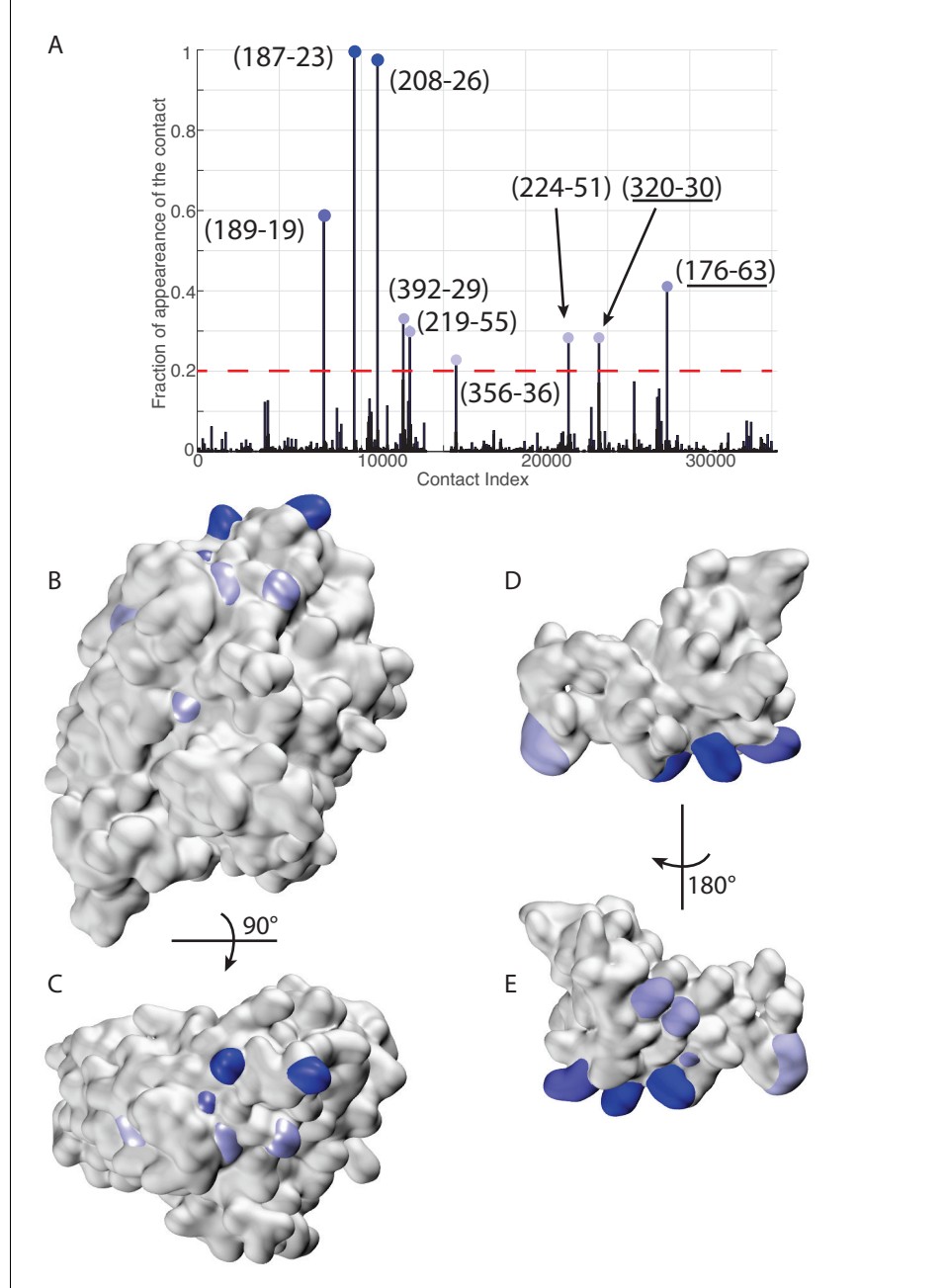

**Figure 7.** Extended analysis of coevolutionary predictions and threshold validation. DCA predicted contacts with threshold selection threshold above 0.2. (A) Frequency of appearance of the coevolutionary inter-protein contacts (see Materials and methods). All nine predicted contacts which appear in more than 20% (dashed red line) of the random matchings are reported. Contacts are colored by their selection frequency (blue: most frequent, white: less frequent). The two contacts containing a buried residue are underlined. (B–C) The seven surface exposed residues among the nine selected, reported on the DnaK NBD. (D–E) The seven surface exposed residues among the nine selected, reported on the JD. Panels B-E follow the same color scheme as panel (A).

The following figure supplement is available for figure 7:

**Figure supplement 1.** Comparison of the five strongest DCA predictions between different methods.

matching procedure. In the context of this work, our interest was restricted to predicting coevolving inter-protein contacts, and we thus applied the simpler random matching strategy discussed above.

## Coevolutionary contact selection

The selection frequencies of all inter-protein contacts were computed over the 1000 realizations (*Figure 3C*) and the most frequently appearing contacts were selected for further analysis. To set a threshold on the number of selected contacts, we computed the solvent accessible surface area (SASA) of the pairs of residues involved in the most frequent contacts. We then selected all ranked contacts before the appearance of a buried residue (SASA < 1 $Å^2$) in the contact pair. This resulted in the selection of three significantly conserved DCA predicted contacts (*Figure 3A,B,C*). Note that similar limited numbers of contacts is generally considered in the DCA prediction of protein-protein interactions (*Hopf et al., 2014*; *Ovchinnikov et al., 2014*).

To validate our conservative threshold choice and further asses the robustness of our results, we extend here the analysis to all DCA predicted contacts with an appearance frequency >20% (*Figure 7*). Among the nine predicted contacts, two involve a buried residue (SASA < 1 Å, *Table 3*) and are discarded from further analysis. The seven remaining DCA predicted inter-protein contacts are depicted in *Figure 7B–D*. We observe that five out of the seven contacts are concentrated in the binding interface observed in the CG simulations that was already identified by the three strongest one and that these additional contacts do not qualitatively change the predictions reported in the results section (*Figure 7*, *Table 3*). This extended analysis of DCA predictions thus confirms the robustness of the results reported in the Results section using a stringent selection criterion, and further supports the DCA predicted binding interface.

## Atomistic simulations

### Atomistic simulations protocol

For all the simulated systems, we used the RosettaDock (*Chaudhury et al., 2011*; *Gray et al., 2003*) protocol to obtain atomistic structures from the low-resolution CG conformations corresponding to the HPD-IN and HPD-OUT binding modes. Particularly, we took advantage of the multi-scale docking protocol (*Chaudhury et al., 2011*) to generate 1000 all-atom conformations from the CG structures corresponding to the center of each HPD-IN and HPD-OUT cluster. We then selected the 10 best scoring structures among those within a deviation equivalent to the radius of the clusters ($C_\alpha$ RMSD $\leq$ 5 Å) and we solvated them in dodecahedral boxes containing approximately 26,000 and 60,000 water molecules for NBD:JD and FL:JD complexes, respectively. MD simulations were

**Table 3.** Solvent accessible surface area (SASA) of the residue involved in the nine predicted DCA contacts with selection frequency higher than 20%. The residue numbering refers to the *E. coli* numbering of DnaK resp. DnaJ. Numbers in parentheses denote the SASA of the amino-acids involved in the contacts normalized to the average amino-acid SASA (data from [*Chothia, 1976*]). Residue pairs involving a buried amino-acid are colored in gray. $\delta$ denotes the average $C_\alpha$ distances of the corresponding residue pairs in FL(ATP) conformations from the CG simulations in the HPD-IN/OUT orientations.

| Contact | Sasa 1 [A$^2$] | Sasa 1 [A$^2$] | $\delta_{OUT}$ [Å] | $\delta_{IN}$ [Å] | Selection frequency |
|---------|-----------|-----------|-----------|-----------|-----------|
| N187 - K23 | 102.7 (0.64) | 119.8 (0.60) | 13.7 | 11.4 | 0.996 |
| D208 - K26 | 41.6 (0.27) | 144.9 (0.72 ) | 8.5 | 10.0 | 0.976 |
| T189 - R19 | 17.3 (0.12) | 177.9 (0.79) | 23.0 | 13.7 | 0.587 |
| A176 - A64 | 0.6 (0.005) | 34.6 (0.30) | - | - | 0.414 |
| L392 - A29 | 89.5 (0.52) | 11.1 (0.10) | 11.7 | 9.7 | 0.334 |
| L219 - E55 | 44.7 (0.26) | 69.8 (0.37) | 22.6 | 17.7 | 0.311 |
| D224 - K51 | 40.7 (0.27) | 95.2 (0.48) | 33.7 | 31.6 | 0.287 |
| L320 - M30 | 0.7 (0.004) | 129.2 (0.70) | - | - | 0.285 |
| F356 - R36 | 48.5 (0.23) | 214.6 (0.95) | 23.3 | 39.9 | 0.232 |

performed using the GROMACS 5 MD package (*Abraham et al., 2015*), with the AMBER14 force-field (*Case et al., 2014*) and TIP3P water model (*Jorgensen et al., 1983*). Given the large internal dynamics of Hsp70, we used harmonic restraints on the backbone atoms of the Hsp70 constructs (NBD(ADP),NBD(ATP) and FL(ATP)) to focus on the inter-protein dynamics in the 30 ns runs. The $\mu$s simulations used the same parameters, with the harmonic restraints removed.

All simulations were performed in a dodecahedral box with periodic boundary conditions. Simulations were carried out with the following protocol:

- Starting structures were solvated with TIP3P water molecules and subsequently energy minimized by steepest descent.
- A first NVT equilibration phase (1 ns) was performed, putting full restraints on all proteins, ATP (when present) and MG atoms.
- A second NPT equilibration phase (1 ns) was performed keeping the same restraints as in the 1ns NVT equilibration phase.
- Subsequently, another NPT equilibration was performed (10 ns), putting restraints on the protein backbone only (DnaK and DnaJ).
- Finally, production runs were carried out for 30 ns, keeping only restraints on the DnaK backbone.
- For the 1$\mu$s simulations, we continued from the last frame of the 30 ns runs, without any restraints.

Temperature was kept constant (T = 300 K) using the v-rescale thermostat (*Bussi et al., 2007*) and NPT (p=1 atm) simulations relied on a Parrinello-Rahman barostat (*Parrinello and Rahman, 1981*). The equations of motion were integrated with a time step of 2 fs. All covalent bonds were constrained to their equilibrium values using the LINCS algorithm (*Hess et al., 1997*). The electrostatic interactions were calculated by the Particle Mesh Ewald algorithm, and a cutoff of 10 nm was used both for Lennard-Jones interaction and for the real-space coulomb contribution.

The distance root mean square (dRMS) measurements were calculated by

$$\text{dRMS(t)} = \sqrt{\frac{1}{N_i N_j} \sum_{(i,j)}^{(N_i,N_j)} (d_{ij}(t) - d_{ij}(0))^2} \tag{5}$$

where $i$ (resp. $j$) are indices of the residues belonging to the J-domain (resp. DnaK), and $d_{ij}(t)$ denotes the distance between the $C_\alpha$ atoms of residue $i$ of the J-domain and residue $j$ of DnaK at time $t$. The dRMSs are then time-averaged over the last 10 ns of the MD trajectories (results reported in *Table 1*).

Similarly the angular stability ($\theta$ in the text, see *Table 1*) was computed by

$$\theta(t) = \frac{180}{\pi} \text{acos}(\vec{I}_1^J(t) \cdot \vec{I}_1^J(0)) \tag{6}$$

where $\vec{I}_1^J$ denotes the inertia axis of the JD associated to the largest moment of inertia computed on the $C_\alpha$ (see *Figure 1—figure supplement 4*), for aligned DnaK NBD at all frames. The values reported in *Table 1* are averaged over the last 10 ns of the MD trajectories.

The binding energies of the JD/DnaK complexes were calculated by the MM-GBSA method, implemented in Ambertools (*Case et al., 2014*). The polar contribution to the solvation energy was calculated using the Generalized Born approximation, with 0.01M counterion concentration in solution. The SASA computation was performed using the LCPO method.

## Acknowledgements

DM thanks the Swiss National Science Foundation (http://www.snf.ch/) for grants 2012_149278 and 20020_163042/1. AJL and GH were supported by the Max Planck Society. AB acknowledges the support of the French Agence Nationale de la Recherche (ANR), under grant ANR-14-ACHN-0016. This work was supported by a grant from the Swiss National Supercomputing Centre (CSCS) under project s684.

# Additional information

## Funding

| Funder | Grant reference number | Author |
|---|---|---|
| Schweizerischer Nationalfonds zur Förderung der Wissenschaftlichen Forschung | 2012_149278 | Duccio Malinverni Paolo De Los Rios |
| Schweizerischer Nationalfonds zur Förderung der Wissenschaftlichen Forschung | 20020_163042/1 | Duccio Malinverni Paolo De Los Rios |
| Max-Planck-Gesellschaft | | Alfredo Jost Lopez Gerhard Hummer |
| Agence Nationale de la Recherche | ANR-14-ACHN-0016 | Alessandro Barducci |

The funders had no role in study design, data collection and interpretation, or the decision to submit the work for publication.

## Author contributions

DM, Conceptualization, Formal analysis, Investigation, Writing—original draft, Writing—review and editing; AJL, Formal analysis, Investigation, Writing—review and editing; PDLR, Conceptualization, Resources, Funding acquisition, Writing—review and editing; GH, Conceptualization, Resources, Formal analysis, Funding acquisition, Writing—review and editing; AB, Conceptualization, Resources, Formal analysis, Supervision, Funding acquisition, Writing—original draft, Writing—review and editing

## Author ORCIDs

Paolo De Los Rios, http://orcid.org/0000-0002-5394-5062
Alessandro Barducci, http://orcid.org/0000-0002-1911-8039

# Additional files

## Supplementary files

• Supplementary file 1. Multiple sequence alignment of the Hsp40 family, in canonical fasta format.

• Supplementary file 2. Organism list of the Hsp40 sequences in *Supplementary file 1*. Each line contains the organism name of the corresponding sequence in the fasta file.

• Supplementary file 3. Multiple sequence alignment of the Hsp70 family, limited to the NBD and inter-domain linker, in canonical fasta format.

• Supplementary file 4. Organism list of the Hsp70 sequences in *Supplementary file 3*. Each line contains the organism name of the corresponding sequence in the fasta file.

• Supplementary file 5. Multiple sequence alignment of the Hsp70 family, full sequences, in canonical fasta format.

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
