## [Decision Letter]

Thank you for submitting your article "Modeling Hsp70/Hsp40 interaction by multi-scale molecular simulations and co-evolutionary sequence analysis" for consideration by *eLife*. Your article has been reviewed by three peer reviewers, and the evaluation has been overseen by a Reviewing Editor and Arup Chakraborty as the Senior Editor. The reviewers have opted to remain anonymous.

Our decision has been reached after consultation between the reviewers. Based on these discussions and the individual reviews below, major revisions are required before the manuscript can be considered further. Please note that there is no guarantee for acceptance.

Summary:

Your paper aims to establish an integrative bioinformatic pipeline, bringing together co-evolutionary modeling with molecular simulations based on both coarse-grained and atomistic models. There is considerable discussion in the field as regards to the relevant interface between DnaK/DnaJ and homologous Hsp70/Hsp40 complexes. Crystal structures and NMR experiments have produced conflicting results. The simulations seem to generally support the conclusions drawn from the NMR studies of the DnaK/DnaJ complex. However, there is no further experimental validation of the computational results, and the study currently lacks new insights into the functional roles of the J protein/Hsp70 interaction. One reviewer commented that she/he could not reproduce the co-evolutionary analyses (although we do not have the MSA). Many other technical points noted below need to be addressed.

Essential revisions:

1) While we find the molecular simulation part very convincing, we are a bit perplexed, possibly due to lack of clarity in the manuscript, about the co-evolutionary analysis. In particular we failed to reproduce some of the results presented. Lacking the multiple sequence alignment (MSA) of the two protein families, we are not sure that we followed the same pipeline as you did. The MSA must be deposited and the pipeline used clearly described. Also, please provide more information about the alignments of the two proteins. How many species are included? What is the statistics of paralogs? How many species with unique copies of both proteins exist in the alignment (how are they correctly matched)? Are there cases of proteins coded in operons or of certified interaction which can be imposed in the matching? You cite Malinverni et al., 2015 for how the MSA was obtained, but this reference does not provide sufficient detail.

2) When you note: "We built two separate seeds containing Hsp70 and Hsp40 sequences, covering a broad portion of the tree of life" did you include sequences other than those for Prokaryotes? If you did so, how could you justify this inclusion as it well known, and also acknowledged by you in the fifth paragraph of the Introduction, that the Bacterial system seems to be incompatible with the Eukaryotic one?

3) Apart from the seed, it is also not clear if eventually non-bacterial sequences were removed from the final MSA. If you did not do so, it would be very important to present the same analysis only on bacterial sequences and discuss differences in the inference (if any).

On a related note, it would be interesting to verify your claim that the statistics over repeated random matchings is informative, by applying your method to PPI treated in past work where the operon-based matching is known. This will provide insights into the generalizability of the approach to other protein systems lacking operons. The same problem has been recently addressed by two papers by Bitbol et al. and by Gueudre et al. in PNAS (2016). Both propose rather involved matching schemes. Would the application of your methods improve the results? In case these papers provide their codes, this would be an easy analysis to be added. The paper has applied the selection criteria of Hopf et al., 2014 with a cutoff of 0.8. The quantity subject to this cutoff is not mentioned. Why should the cutoff established for operonic matchings be applicable to random matchings? We are rather surprised (positively) that a random matching produces similarly strong signal.

4) The method used to match the two protein families is interesting but recently two other methods have been published tackling the issue of concatenating MSAs with a new computational approach: Thomas Gueudré et al. "Simultaneous identification of specifically interacting paralogs and interprotein contacts by direct coupling analysis", vol. 113 no. 43, 12186-12191, doi:10.1073/pnas.1607570113, Published online before print October 11, 2016. Anne-Florence Bitbol, Robert S. Dwyer, Lucy J. Colwell, and Ned S. Wingreen Inferring interaction partners from protein sequences PNAS 2016 113 (43) 12180-12185; published ahead of print September 23, 2016, doi:10.1073/pnas.1606762113. We encourage you to try one or both of the methods mentioned to verify the robustness of the random matching.

5) Together with the selection criterion introduced in Hopf et al., 2014, it could be interesting to show in general a histogram across the 1000 stochastically concatenated MSAs of the original residue-residue score (Ekeberg, Hartonen and Aurell, 2014) in order to figure out whether the score is doing more than largely producing top-scoring pairs. Similarly, how many protein pairs typically have a score larger than 0.8?

6) The long all-atom MD runs are only performed for the HPD-IN conformation. Would runs on the HPD-OUT conformation confirm the preference for the IN conformation over the OUT conformation?

7) A critically important overarching question is what have we learned from this study that was not known previously? The interaction site on DnaK is not unexpected, the region on the J-domain, especially the HPD, was known to be key to this interaction, and the dynamic nature of the complex was expected. Thus, the work, while laudable, in its current form, does not move the field significantly forward. You need to attempt to address some of the functional underlying questions: how do J-proteins modulate Hsp70s to affect their allosteric cycle? What is the role of the diversity of J-proteins and how involved is the SBD?

[Editors' note: further revisions were requested prior to acceptance, as described below.]

Thank you for resubmitting your work entitled "Modeling Hsp70/Hsp40 interaction by multi-scale molecular simulations and co-evolutionary sequence analysis" for further consideration at *eLife*. Your revised article has been favorably evaluated by Arup Chakraborty (Senior Editor), a Reviewing Editor, and three reviewers.

The manuscript has been improved but there are some remaining issues that need to be addressed before acceptance, as outlined below:

1) In previous literature (namely Feinauer et al., 2016; Gueudré et al., 2016), an empirical coevolutionary score for protein pairs was introduced: starting from the Average Product Corrected (APC) inter-protein residue pair score (i.e. the restriction of the APC coupling score to all pair of residues i, j for which i belongs to one protein and j to the second one), and consider the mean over the 4 largest. It would be extremely interesting to show this score for the random matching strategy and also for PPM and IPA to compare the "strength" the DnaJ/DnaK coupling in comparison with other known protein pairs presented in literature.

2) While it becomes clear from the manuscript and from the authors reply that the central interest is in the Hsp70/Hsp40 system and not in the development of a general-purpose methodology, two results reported in the rebuttal letter but not in the manuscript or its supplement should be reported in the supplement, with a short reference from the new last paragraph of the subsection “3. Random Paralog matching”:

The random matching procedure is successful even if applied to the two component system used in Bitbol et al., 2016 and Gueudré et al., 2016.

The matching procedure of Bitbol et al., 2016 and Gueudré et al., 2016 produces strongly overlapping results with the random-matching procedure.

3) A minor remark concerns the results of the 3701 always matched protein pairs, where both sequences are single copy in their genomes. It is interesting that this case recovers part of the strongest signal, but the sampling of random matchings is able to enhance the coevolutionary signal beyond the one found for the uniquely matched pairs. Again, this is a nice detail, it might be introduced at the beginning of the Methods section on random matching, or at the end of Sec. II.B. “Coevolutionary Analysis predicts conserved DnaK-DnaJ contacts”.

4) HMMer should be cited in the first paragraph of the subsection “1. Sequence Extraction and Preprocessing”.

---

## [Author Response]

*Essential revisions:*

*1) While we find the molecular simulation part very convincing, we are a bit perplexed, possibly due to lack of clarity in the manuscript, about the co-evolutionary analysis. In particular we failed to reproduce some of the results presented. Lacking the multiple sequence alignment (MSA) of the two protein families, we are not sure that we followed the same pipeline as you did. The MSA must be deposited and the pipeline used clearly described. Also, please provide more information about the alignments of the two proteins. How many species are included? What is the statistics of paralogs? How many species with unique copies of both proteins exist in the alignment (how are they correctly matched)? Are there cases of proteins coded in operons or of certified interaction which can be imposed in the matching? You cite Malinverni et al., 2015 for how the MSA was obtained, but this reference does not provide sufficient detail.*

We agree with the reviewers that the protocol of sequence extraction and the coevolutionary analysis need further explanations. As requested, the multiple sequence alignments of the two protein families, as well as the taxonomic identifiers of all sequences, have now been uploaded as supplementary material available online. Furthermore, we added a more detailed presentation of the sequence extraction and preprocessing steps in the Materials and methods section and we provided details about the statistics of the multiple sequence alignments. We report here for convenience in Table 4 the number of sequences and organisms in the alignments used in this work.

Author response table 1.Summary of the taxonomic composition of the two alignments used in the work. Entries of the table represent the number of sequences found in each taxonomic group. The number of organisms in each taxonomic group is indicated in parenthesis.**DOI:**
http://dx.doi.org/10.7554/eLife.23471.030EukaryotesBacteriaArchaeaVirusesOtherTotalHsp4014369 (1093)11379 (7837)311 (273)36 (22)159 (13)26254 (9238)Hsp707881 (1933)11819 (8272)273 (258)25 (17)63 (13)20061 (10493)

Author response image 1.Distribution of number of paralogs per organism for Hsp40 and Hsp70.**DOI:**
http://dx.doi.org/10.7554/eLife.23471.031

The number of Hsp40 and Hsp70 paralogs per organism varies widely. There are in total 8856 organisms for which both proteins have been retrieved. Among these, 3701 organisms have a single (retrieved) copy of both Hsp40 and Hsp70. These single sequence pairs were added to the matched MSAs without any special treatment. To assert the influence of the single copy pairs on the coevolutionary results, we performed DCA on an MSA composed solely of the 3701 organisms having a single copy of Hsp40 and Hsp70. Note that due to the finite sensitivity of the sequence extraction protocol, there is no guarantee that these organisms have truly a unique pair of Hsp40/Hsp70. The two strongest DCA contacts predicted on this reduced set (Table 5) correspond to the two most frequent contacts in the complete set. We however observed no overlap between lower-rank predicted inter-protein contacts in this reduced set and other frequently selected contacts in the full set. These observations lead to two conclusions: First, the organisms containing single pairs positively contribute to the most predicted contacts in the full dataset. Second, not the entire coevolutionary signal detected in the full dataset is contained in the reduced dataset containing only single pair organisms. The combination of both organisms with single and multiple paralogs is thus beneficial for the overall prediction of inter-protein contacts.

Author response table 2.10 first DCA predicted contacts using the reduced set consisting of the 3701 single copy pairs organisms. The second column denotes the DCA score of the predicted contact. The rank is computed over all contacts, including the intra-chain predictions.**DOI:**
http://dx.doi.org/10.7554/eLife.23471.032ContactScoreRankSASA [Å2]N187-K230.31324102.7 – 119.8D208-K260.2538542.0 – 145.0A17-R630.253920 – 84.0E306-R630.2440641.1 – 84.0I338-I210.234480.8 – 1.0N222-K310.2246314.8 – 145.0L219-E550.2247844.7 – 69.8L390-A290.2057889.5 – 11.1T215-E550.1959726.6 – 69.8L131-A640.1961747.2 – 34.6

There are cases of Hsp40s and Hsp70s that appear in operons in bacterial organisms (e.g. members of the DnaJ/DnaK and HscB/HscA subfamilies). We initially tested the operon based matching presented in (Ovchinnikov et al., *eLife* 2014), where sequences are matched if they are separated by less than 20 genes on the genome. Although a substantial number of matched pairs are retrieved (11293), we observed that these pairs of sequences present very high sequence identity, which is reflected by the effective number of sequences they represent (N_eff_=1045, with similarity threshold of maximum 90% sequence identity). This was reflected by overall poor DCA predictions, and particularly weak inter-protein couplings. Furthermore, there are some known interacting paralogs for a very small set of model organisms. However, these sets are not exhaustive (they do not cover all paralogs of the model organisms, even in the case of *E. coli*).

Given the reasons above, and the quality of the inter-protein predictions based on the random matching strategy, we decided to keep the pairing procedure as generic and stochastic as possible, without introducing any additional pairing restraints.

*2) When you note: "We built two separate seeds containing Hsp70 and Hsp40 sequences, covering a broad portion of the tree of life" did you include sequences other than those for Prokaryotes? If you did so, how could you justify this inclusion as it well known, and also acknowledged by you in the fifth paragraph of the Introduction, that the Bacterial system seems to be incompatible with the Eukaryotic one?*

We thank the reviewers for pointing to the importance of the possible differences between Bacteria and Eukaryotes regarding the Hsp40-Hsp70 interactions. As discussed above (see point 1), no sequences were discarded based on taxonomic information. We kept sequences from all taxonomic origins in our main dataset to incorporate as much variability and exhaustiveness as possible in our dataset. We repeated the analysis on bacterial sequences only (200 random matchings). For completeness, we also performed the same analysis on eukaryotic sequences only (200 random matchings). Results of the coevolutionary analysis restricted to bacterial and eukaryotic organisms separately show that the origin of the inter-protein contacts detected by DCA in the full alignment mostly stems from the bacterial sequences (Figure 9, left) while no strong coevolutionary signal is retrieved from eukaryotic sequences alone (Figure 9, right). The contacts extracted from bacterial data alone are in excellent agreement with the predictions from the full dataset (Table 6). This last observation highlights the robustness of the matching procedure, as the addition of sequences from eukaryotic organisms does not perturb the ranking of the strongest coevolving DCA contacts.

Author response image 2.Frequency of appearance of coevolutionary inter-protein contacts.Left: Only bacterial sequences. Right: Only eukaryotic sequences. Inset: Zoom on the vertical axis between 0 and 5%. The five most frequent contacts in the bacterial analysis reported in Tab.R3 are highlighted. The colored circles coincide with the three most frequently selected contacts in the full dataset, reported in the main text.**DOI:**
http://dx.doi.org/10.7554/eLife.23471.033

The five most frequent contacts in the bacterial analysis (Figure 9, left) are reported in Table 6. We observe that the four first contacts (denoted by (*) in the table) are among the first five contacts predicted in the complete dataset. These observations support the idea that the coevolutionary signal that we observe in the complete dataset stems from the bacterial sequences in our sample. We added these results, based on separate bacterial and eukaryotic sequences to the manuscript (Figure 3—figure supplement 1) and commented on them in the main text.

Author response table 3.Five most frequent contacts in the coevolutionary analysis restricted to bacterial sequences. Contacts denoted by (*) are among the five first predicted contacts in the full dataset.**DOI:**
http://dx.doi.org/10.7554/eLife.23471.034ContactFrequencyD208 – K26 (*)1N187 – K23 (*)1T189 – R19 (*)0.83L392 – A29 (*)0.81V215 – Y540.75

*3) Apart from the seed, it is also not clear if eventually non-bacterial sequences were removed from the final MSA. If you did not do so, it would be very important to present the same analysis only on bacterial sequences and discuss differences in the inference (if any).*

*On a related note, it would be interesting to verify your claim that the statistics over repeated random matchings is informative, by applying your method to PPI treated in past work where the operon-based matching is known. This will provide insights into the generalizability of the approach to other protein systems lacking operons. The same problem has been recently addressed by two papers by Bitbol et al. and by Gueudre et al. in PNAS (2016). Both propose rather involved matching schemes. Would the application of your methods improve the results? In case these papers provide their codes, this would be an easy analysis to be added. The paper has applied the selection criteria of Hopf et al., 2014 with a cutoff of 0.8. The quantity subject to this cutoff is not mentioned. Why should the cutoff established for operonic matchings be applicable to random matchings? We are rather surprised (positively) that a random matching produces similarly strong signal.*

We addressed above (point 2) the question regarding the consequences of limiting the DCA analysis to bacterial (or non-bacterial) sequences. For the sake of clarity, we discuss here the application of our random matching algorithm to an alternative system as requested by the reviewers. Please find below our considerations about the application of the IPA (Iterative Pairing Algorithm, IPA, Bitbol et al.) and PPM (Progressive Paralog Matching, PPM, Guedré et al.) schemes to the Hsp40/Hsp70 case (point 4) and the selection criteria (point 5).

We want to stress that the scope of our work is to investigate Hsp70/Hsp40 interactions and not to propose a novel, general approach to investigate PPIs where operon based matching is not available. However, we are pleased by the fact that the reviewers are positively surprised by the results obtained with the random matching algorithm proposed here and we tried to address their curiosity about its generalizability to other systems. Following their suggestion, we assessed the robustness of the random matching strategy by performing the same analysis on the bacterial two-component system HK-RR, which was used by both Guedré et al., and by Bitbol et al. as a benchmark and validation set. Unfortunately, we could not access the MSAs used in these two studies that apparently were not made publicly available. We then reconstructed them following the procedure for the data extraction and preprocessing described in the work by Bitbol et al. and briefly summarized here:

Two component systems sequences were retrieved from the P2CS database. We retained sequences belonging to the Pfam domain profiles of the Histidine Kinase (HisKA, PF00512) or the Response Regulator (Response_reg, PF00072). After removing organisms containing a single pair of HK/RR and removing hybrid and unorthodox sequences, the interacting partners were retrieved as adjacent HK and RR on the genome. Finally, a reduced dataset is under-sampled, by extracting randomly 459 organisms (see Bitbol et al.). All organisms possessing a single pair of HK-RR are discarded. We verified that the average number of paralogs and the distribution of number of paralogs match with those presented in their work.

We report in Figure 10 the results obtained with the random matching strategy (1000 random matchings) on the resulting HK/RR dataset. Interestingly, we observe the presence of a limited set of predicted contacts that are observed very frequently (Figure 10, left). This is particularly noteworthy in the case of the HK/RR dataset, which has a broader distribution of paralogs per organism compared to our dataset of Hsp40/Hsp70, and thus a higher variability in the matchings. We also notice that the most frequent couplings (Figure 10, right) either correspond to native contacts in the crystal structure (PDB: 3DGE) (5 out of the 10 most frequent), or lie very close to native contacts. The random matching strategy is thus generalizable to the bacterial HK-RR system for coevolutionary inter-protein contact predictions.

Author response image 3.Random matching results obtained on the HK/RR dataset.Left: Frequence of appearances of the contacts. The 15 most frequent contacts are highlighted by red circles for visualization. Right: The 10 most frequent contacts plotted onto the contact map of the HK/RR hetero-dimer (PDB:3DGE). Structural contacts from the PDB are defined using a threshold of 8.5Å between heavy atoms of two residues. Green (resp. red) DCA predicted contacts correspond to true (resp. false) predictions with respect the crystal structure. The blue dotted line represents the limit between HK and RR.**DOI:**
http://dx.doi.org/10.7554/eLife.23471.035

We want to acknowledge here that both IPA and PPM tackle an ambitious objective, namely the simultaneous identification of both the coevolving residues forming the interaction interface and the interaction network between multiple paralogs. Both studies propose technically involved and elegant numerical solutions for tackling these coupled problems. In our case, the objective was restricted to determining a set of coevolving residues between Hsp40 and Hsp70. While of crucial importance and great interest, the identification of the interaction network between Hsp40 and Hsp70 goes beyond the scope of this work, and will probably require a more detailed analysis of the effect of highly promiscuous interactions on coevolving interfaces.

*4) The method used to match the two protein families is interesting but recently two other methods have been published tackling the issue of concatenating MSAs with a new computational approach: Thomas Gueudré et al. "Simultaneous identification of specifically interacting paralogs and interprotein contacts by direct coupling analysis", vol. 113 no. 43, 12186-12191, doi:10.1073/pnas.1607570113, Published online before print October 11, 2016. Anne-Florence Bitbol, Robert S. Dwyer, Lucy J. Colwell, and Ned S. Wingreen Inferring interaction partners from protein sequences PNAS 2016 113 (43) 12180-12185; published ahead of print September 23, 2016, doi:10.1073/pnas.1606762113. We encourage you to try one or both of the methods mentioned to verify the robustness of the random matching.*

We thank the reviewers for highlighting the two recent methods proposed for the matching of interacting partners in the presence of paralogs.

In order to assess the robustness of our results with respect to multiple methods, we performed two additional analyses, using the PPM algorithm presented in Guedré et al., and the IPA algorithm presented in Bitbol et al. We implemented both IPA and PPM in Matlab (from scratch for IPA, based on the Julia implementation and pseudo-code provided for PPM). Both algorithms had to be adapted to allow for an unequal number of paralogs of both families. In the case of IPA, we started from a random pairing seed as presented in their original work. Both algorithms were run with their default parameters as presented in their resp. papers. To best compare the contact prediction quality, we used IPA/PPM to build (near-) optimal paralog matchings. The resulting MSAs were then used as input to the same pseudo-likelihood based DCA algorithm used in the random matching strategy, which is known to perform slightly better than the mean-field methods used in IPA/PPM when performing contact predictions.

We report in Table 7 the nine most frequent contacts in the random matching approach, as well as the nine strongest coevolving contacts in the DCAs obtained from the MSAs resulting from the IPA/PPM procedures. We highlighted in green the contacts that appear in both the Random matching and either the IPA or the PPM procedures. In yellow are contacts that are defined as “similar” (one residue being predicted by two methods and the second one in a nearby region between the two predictions). As seen, the results obtained by the random matching strategy are strongly overlapping with the results obtained by IPA and PPM. In particular, the four most frequent contacts identified by the random matching are also among the strongest contacts identified by the other methods. The 5^th^ and 6^th^ most frequent contacts in the random matching are similar (see above) to strong contacts identified by IPA or PPM. Taken together, these results highlight the strong overlap of all three methods in this case.

Author response table 4.Comparison of top inter-protein contact predictions between Random Matching, PPM and IPA. In green are highlighted the overlapping predictions between Random matching and PPM/IPA. Contacts highlighted in yellow denote similar contacts, defined as having one identical residue, and the second one in proximity.**DOI:**
http://dx.doi.org/10.7554/eLife.23471.036**Random Matching****PPM****IPA**K23 – N187K23 – N187K23 – N187K26 – D208E55 – T215R19 – T189R19 – T189K26 – D208K26 – D208A64 – A176R63 – A17D59 – I39A29 – L392R19 – T189A64 – A176E55 – L219Y25 – A191A29 – L382K51 – D224A29 – L382K50 – Y193M30 – L320Y25 – I338A24 – G358R36 – F356Y54 – A376I21 – I338

To further investigate whether IPA or PPM could improve the predictions on the bacterial/eukaryotic subsets, we repeated the analysis on these two separate subsets. Our results (Figure 11) are in perfect agreement with our observations obtained using the random matching strategy (reported again in Figure 11 for visual comparison, top row). We observe a strong overlap between all methods in the case of the full dataset and the dataset restricted to bacterial sequences (left and middle column), and a large spread of predictions over three different lobes with all methods in the case of the dataset restricted to eukaryotic sequences (right column).

Author response image 4.Comparison of inter-protein predictions between the three methods, on the full dataset (left column), bacterial dataset (central column) and eukaryotic dataset (right column).The residues on the NBD involved in the absolute top five contacts predicted by all methods are reported, without any filtering. The color-code denotes the rank of the predictions, from dark blue (rank=1) to dark red (rank=5).**DOI:**
http://dx.doi.org/10.7554/eLife.23471.037

Altogether, given the strong agreement for the top predictions between the three methods, we can conclude that the random matching strategy used in this work produces results fully compatible with the two state-of-the art methods. Furthermore, we observe that the use of more involved methods currently does not allow disentangling the more complex eukaryotic coevolutionary networks between paralogs. Indeed, both IPA and PPM were developed and benchmarked on datasets with fundamentally binary interactions, i.e. the underlying assumption of both methods is to find a unique one-to-one matching for all paralogs. While this is perfectly legit and documented for some systems, the situation for the Hsp40-Hsp70 interaction network is notoriously more complex. Several cases of promiscuous interactions of Hsp70s with multiple different J-proteins have been experimentally reported. The promiscuous one-to-many, and possible many-to-many, interactions in this system is highlighted by the unbalanced distribution of paralogs per organism between Hsp70 and Hsp40 (e.g. 15 Hsp70 vs 50 Hsp40 in H. Sapiens). The approaches consisting in finding the optimal one-to-one matching, as implemented in IPA and PPM is therefore not guaranteed to be an optimal solution for more complex interaction networks such as Hsp40 and Hsp70.

*5) Together with the selection criterion introduced in Hopf et al., 2014, it could be interesting to show in general a histogram across the 1000 stochastically concatenated MSAs of the original residue-residue score (Ekeberg, Hartonen and Aurell, 2014) in order to figure out whether the score is doing more than largely producing top-scoring pairs. Similarly, how many protein pairs typically have a score larger than 0.8?*

We agree with the reviewers that the selection criterion used needs further clarifications. We extensively discussed in detail the definition and the meaning of the selection threshold used.

As discussed in the revised manuscript, the selection score pertains to the selection of inter-protein residue pairs for a given MSA realization. In practice, all inter-protein DCA scores are normalized (see Equation 4 of the revised manuscript), and a threshold is set on the normalized scores to select the significant predicted contacts for each random realization of the random matching. It must be noted that the normalization does not change the relative rank of the inter-protein contacts, as it is merely a convenient way to rescale the scores, which can in general vary with the depth and width of the MSAs across different protein families. Alternatively, one could set a threshold on the number of highest scoring contacts selected for each realization, which leads to virtually the same results.

*6) The long all-atom MD runs are only performed for the HPD-IN conformation. Would runs on the HPD-OUT conformation confirm the preference for the IN conformation over the OUT conformation?*

We would like to underline that the long all-atom MD simulations were not performed to discriminate between the alternative HPD-IN and HPD-OUT conformations. Their objective was to further assess the stability of the proposed model on longer time-scales. We selected the HPD-IN conformation as a model for the DnaK/DnaJ complex due to its better agreement both with the coevolutionary predictions and with previous experimental evidences, such as the involvement of the HPD tripeptide in the stimulation of ATP hydrolysis by Hsp70.

However, we followed the suggestion of the reviewers and we further investigated the stability of the HPD-OUT conformations of the full-length complex using longer explicit-solvent atomistic MD runs. To this aim, we selected the two most stable HPD-OUT conformations in the 30ns runs and simulated them on longer time-scales using the same protocol as for the long HPD-IN simulations. Given the considerable computational cost of these simulations (approximatively 200’000 atoms), we could perform two ~750 ns MD runs for the HPD-OUT conformations.

In this timescale, we did not observe the full detachment of the J-domain from the docked state. This outcome is not totally surprising even in the case of weak, non-specific interactions as it is well known that the current force-fields have a propensity to over-estimate the stickiness of proteins and under-estimate the solvent-protein interactions (Petrov and Zagrovic, PLoS Comp. Biol, 2014, Abriata and Dal Peraro, Sci. Rep., 2015). Therefore, we relied on a standard, implicit solvation approach (MM-GBSA) to estimate the differential stability of the HPD-IN and HPD-OUT complex from the atomistic MD trajectories. The results, shown in Figure 12, indicate a striking difference in the inter-protein binding enthalpy between the HPD-IN and HPD-out trajectories. Although this analysis does not account for entropic differences, the observed difference clearly supports a stronger binding in the HPD-IN conformation.

Author response image 5.Binding energy computed by GBSA.The blue bars denote the three micro-second runs of the HPD-IN conformations. The red bars denote the two 750 ns runs of the HPD-OUT conformations.**DOI:**
http://dx.doi.org/10.7554/eLife.23471.038

*7) A critically important overarching question is what have we learned from this study that was not known previously? The interaction site on DnaK is not unexpected, the region on the J-domain, especially the HPD, was known to be key to this interaction, and the dynamic nature of the complex was expected. Thus, the work, while laudable, in its current form, does not move the field significantly forward. You need to attempt to address some of the functional underlying questions: how do J-proteins modulate Hsp70s to affect their allosteric cycle? What is the role of the diversity of J-proteins and how involved is the SBD?*

We respectfully disagree with the reviewers on this point. Although it is true that there are multiple experimental evidences about the interaction between J-proteins and Hsp70s, only two structural models are currently available to rationalize these observations: the X-ray structure of a cross-linked eukaryotic NBD-JD complex (Jiang et al., 2007) and a model for the interaction between DnaJ JD with ADP-bound DnaK based on PRE-NMR data (Ahmad et al. 2011). While both these studies are insightful, they clearly suffer from some limitations, as discussed in the manuscript. To date no structural model is available for the complex formed by DnaJ and ATP-bound DnaKi.e. for the most studied Hsp40/Hsp70 system in the functionally-relevant nucleotide state (as stated for example in the recent review by Zuiderweg et al., Cell Stress and Chaperones, February 2017, section “Complexes with J proteins, p.182-182). Furthermore, a detailed structural characterization of the Hsp70/Hsp40 complex is critical both for better understanding interactions with other chaperone families, such as Hsp90s (Kravats et al.,2016, 429 (6), p. 858 J Mol Biol) and for designing allosteric inhibitors (Li et al., 2016, 16 (25), p. 2729, Curr Top Med Chem). Here, we propose a novel structural model for the transient DnaK/DnaJ complex that is i) in excellent agreement with co-evolutionary data, and thus expected to be functionally-relevant and ii) compatible with the main experimental observations available in the literature at variance with previous models (Ahmad et al. 2011). We acknowledge that these points have probably not been satisfactorily explained in the manuscript and we modified the revised version to address this issue.

Importantly, we were inspired by the insightful comments of the reviewers and we thus extended our work to investigate the involvement of SBD in J-protein binding. Moreover, we took advantage of our findings to propose a mechanistic hypothesis about the role of J-protein binding in Hsp70 functional cycle. Particularly, we calculated the energetic contribution of individual residues to the binding interface for elucidating the structural determinants of the dynamical DnaK/JD interactions. This analysis assessed the importance of the interdomain linker and neighbouring NBD residues for effective JD binding and also unveiled a potential role of SBD in the stabilization of the complex. The relevance of SBD contribution was then further confirmed by an additional co-evolutionary analysis taking into account full-length Hsp70 sequences. Combining our findings and recent results on the allosteric signal transmission in DnaK, we then formulated an hypothesis about the molecular mechanism at the basis of JD-induced stimulation of DnaK ATPase activity. Indeed, our structural, coevolutionary and energetic analyses suggest a scenario where JD binding shifts the equilibrium of ATP-bound DnaK, which dynamically populates multiple conformational states, toward “allosterically active” conformers with the highest ATPase activity.

[Editors' note: further revisions were requested prior to acceptance, as described below.]

*The manuscript has been improved but there are some remaining issues that need to be addressed before acceptance, as outlined below:*

*1) In previous literature (namely Feinauer et al., 2016; Gueudré et al., 2016), an empirical coevolutionary score for protein pairs was introduced: starting from the Average Product Corrected (APC) inter-protein residue pair score (i.e. the restriction of the APC coupling score to all pair of residues i, j for which i belongs to one protein and j to the second one), and consider the mean over the 4 largest. It would be extremely interesting to show this score for the random matching strategy and also for PPM and IPA to compare the "strength" the DnaJ/DnaK coupling in comparison with other known protein pairs presented in literature.*

We now reported in the manuscript the inter-protein scores introduced in Feinauer et al., 2016 and Gueudré et al., 2016 obtained for Hsp40-Hsp70 using the random matching, IPA and PPM. Nevertheless, we note that as the random matching results in an ensemble of MSAs, we could only report the empirical score averaged over the 1000 random realizations.

*2) While it becomes clear from the manuscript and from the authors reply that the central interest is in the Hsp70/Hsp40 system and not in the development of a general-purpose methodology, two results reported in the rebuttal letter but not in the manuscript or its supplement should be reported in the supplement, with a short reference from the new last paragraph of the subsection “3. Random Paralog matching”:*

The random matching procedure is successful even if applied to the two component system used in Bitbol et al., 2016 and Gueudré et al., 2016.

*The matching procedure of Bitbol et al., 2016 and Gueudré et al., 2016 produces strongly overlapping results with the random-matching procedure.*

As suggested by the reviewers, we included and discussed the comparison of the results obtained with IPA, PPM and random matching approaches for the Hsp70/Hsp40 families.

As discussed with the editorial board, we decided not to discuss in the main manuscript the positive results obtained on the unrelated HK/RR dataset.

3) A minor remark concerns the results of the 3701 always matched protein pairs, where both sequences are single copy in their genomes. It is interesting that this case recovers part of the strongest signal, but the sampling of random matchings is able to enhance the coevolutionary signal beyond the one found for the uniquely matched pairs. Again, this is a nice detail, it might be introduced at the beginning of the Methods section on random matching, or at the end of Sec. II.B. “Coevolutionary Analysis predicts conserved DnaK-DnaJ contacts”.

We discussed the results on the separate dataset containing only the 3701 single-copy pairs in the random matching sub-section of the Methods section.

*4) HMMer should be cited in the first paragraph of the subsection “1. Sequence Extraction and Preprocessing”.*

We apology for our oversight of the citation of the HMMER software suite and added the citation to the revised manuscript.